# Comparison of the Basal Cell Carcinoma (BCC) Tumour Microenvironment to Other Solid Malignancies

**DOI:** 10.3390/cancers15010305

**Published:** 2023-01-02

**Authors:** Eliana-Ruobing Zhang, Sarah Ghezelbash, Pingxing Xie, Misha Fotovati, Ivan V. Litvinov, Philippe Lefrançois

**Affiliations:** 1Faculty of Medicine, McGill University, Montreal, QC H3T 1E2, Canada; 2Division of Experimental Medicine, Department of Medicine, McGill University, Montreal, QC H3T 1E2, Canada; 3Lady Davis Institute for Medical Research, Montreal, QC H3T 1E2, Canada; 4Division of Dermatology, Department of Medicine, McGill University, Montreal, QC H3T 1E2, Canada; 5Division of Dermatology, Department of Medicine, Jewish General Hospital, Montreal, QC H3T 1E2, Canada

**Keywords:** basal cell carcinoma, BCC, genomics, computational biology, tumour microenvironment, CIBERSORT, xCell, TCGA, The Cancer Genome Atlas

## Abstract

**Simple Summary:**

Basal cell carcinoma is the most common human cancer. Most BCCs are low-risk and are easily treated; however, 1–2% are aggressive and highly destructive to the surrounding skin, called advanced BCC. It was discovered that this subtype has a different immune profile than routine BCC, and they contain a special type of stem cell population that helps them grow and spread. There is currently no reliable laboratory model for advanced BCC, making it hard to further study it and find new treatments. For these reasons, this project was conducted using genomic data from 11,000 tumours coming from 33 non-BCC cancer types. Using computational biology, we have compared the immune cell makeup of the tumour microenvironments to determine the top three most similar cancers, which we will call BCC “relatives”. We will examine how these “relatives” develop and grow, as well as current existing treatments and their response to such treatments.

**Abstract:**

Basal cell carcinoma (BCC) is the most common form of skin cancer, contributing to nearly a third of new cancer cases in Western countries. Most BCCs are considered low risk “routine” lesions that can either be excised through surgery or treated with chemotherapeutic agents. However, around 1–2% of BCC cases are locally aggressive, present a high risk of metastasis, and often develop chemoresistance, termed advanced BCC. There currently exists no animal model or cell line that can recapitulate advanced BCC, let alone intermediate-risk and high-risk early BCC. We previously found that aggressive BCC tumours presented a Th2 cytokine inflammation profile, mesenchymal stem cell properties, and macrophage-induced tumoral inflammation. In this study, we aimed to identify potential BCC “relatives” among solid-organ malignancies who present similar immune cell proportions in their microenvironment compositions. Using immune cell type deconvolution by CIBERSORTx, and cell type enrichment by xCell, we determined three cancers with the most similar tumour microenvironments as compared to BCC. Specifically, chromophobe renal cell carcinoma, sarcoma, and skin cutaneous melanoma presented significance in multiple cell types, namely in CD4+ T lymphocytes, gammadelta T lymphocytes, and NK cell populations. Consequently, further literature analysis was conducted to understand similarities between BCC and its “relatives”, as well as investigating novel treatment targets. By identifying cancers most like BCC, we hope to propose prospective druggable pathways, as well as to gain insight on developing a reliable animal or cell line model to represent advanced BCC.

## 1. Introduction

Basal cell carcinoma (BCC) is the most common type of skin cancer and accounts for roughly 80% of all skin cancers, with increased risk in older, fair-skinned individuals [1,2]. In particular, the risk of Caucasians being diagnosed with BCC reaches up to 40% [3]. Major risk factors include age, UV exposure (particularly UVB), and immunosuppression. Current understanding of BCC progression and tumorigenesis revolves around the Sonic hedgehog (Shh) pathway, an important regulator of cell differentiation and tumorigenesis [4,5]. Its constitutive activation, specifically through *PTCH1* inactivation [6], has been shown to be crucial for both sporadic and familial forms of BCC [7]. Small molecule inhibitors such as vismodegib and sonidegib, which target SMO (Smoothened) in the Shh pathway, have been FDA-approved for treatment of advanced BCC [8]; however, they are limited by their severe side effects and failure of complete patient response [9].

Although most BCCs are considered “routine”, or low-risk, and are simple to treat, around 1–2% of BCC cases are locally aggressive and present a high risk of metastasis [10]. These advanced cases are often resistant to the targeted therapies mentioned above. Currently, there exists no reliable lab models in either animal or cell lines to study advanced BCC, owing to their lack of important features in experimental models [11], or their difficulty of manipulation [12], respectively. We recently determined that, in comparison to non-advanced BCC, aggressive BCC tumours presented an inflammatory cytokine profile that shifted to a Th2 cytokine inflammation profile, which is thought to be more tumour permissive [13] and may include pathways such as Toll-like receptor, PDGFR, and extracellular matrix remodeling [14]. Furthermore, advanced BCC had a higher enrichment for mesenchymal stem cells [13]. 

Given the lack of reliable experimental models for BCC, especially more aggressive, higher risk, or advanced disease, we aimed to use our findings related to tumour microenvironment to identify other cancers/cancer subtypes most resembling BCC. Hence, this manuscript aims to identify non-BCC cancers that present comparable tumour microenvironment features to that of BCC (“relatives”), in hopes to contribute to both the development of a reliable model for advanced BCC and novel potential therapeutic avenues.

## 2. Materials and Methods

### 2.1. Data Acquisition and RNA-Seq Processing

Whole-genome RNA-sequencing data for 75 BCC samples was obtained and processed as previously described [13,14]. Non-BCC whole-genome RNA-sequencing data was extracted from all 33 cancer types in The Cancer Genome Atlas consortium, and processed as similarly described [15].

### 2.2. Cell-Type Enumeration Using RNA Deconvolution

To identify the closest BCC “relatives”, RNA deconvolution in CIBERSORTx was performed, using the standard LM22 leukocyte signature matrix obtained from 22 pure immune cell lines [16] and 100 permutations, to estimate the relative number of each cell subtype. The following analyses were performed: all B cells, all CD4+ T cells, CD8+ T cells, Treg cells, all NK T cells, T γδ (gamma delta) cells, total lymphocytes, and total macrophages, as previously performed [17]. Primary analyses were conducted using CIBERSORTx’s standard relative score. xCell analysis was also performed to determine cell type abundance scores using xCell’s standard 64 cell type signatures [18]. To consider relatedness of cancers, we tested, for a specific feature, whether the distributions of scores for that feature were similar between BCC and the non-BCC malignancy. We considered non-rejection of the null hypothesis using asymptotic two-sample Kolmogorov–Smirnov test to compare and validate significant distributions. The top three (if available) cancers with the highest *p*-values over 0.05 were considered as a potential BCC “relative”. CIBERSORTx and xCell scores were separately Bonferroni-corrected for multiple hypothesis testing. Figures were generated using Graphpad Prism (violin plots) and Rtsne R package (tSNE plot).

## 3. Results

We compared the microenvironment of 75 BCC samples to over 11,000 tumours across 33 cancer subtypes from the TGCA using xCell (Figure 1) and CIBERSORTx (Figure 2), with immune cell types selected according to previous findings [13]. Using these guiding cell type populations, we identified the top three most recurring cancers. Chromophobe renal cell carcinoma (KICH) presented similar distributions in its scores for CD4+ cells (*p* = 0.17), macrophages (*p* = 0.4904), Th2 cells (*p* = 0.151), and Tgammadelta cells (*p* = 0.5525). Sarcoma (SARC) showed significant *p*-values in its total lymphocyte (*p* = 0.2290), NK cells (*p* = 0.1979), Tgammadelta cells (*p* = 0.07291), and CD4+ (*p* = 0.6344) cell populations. Bladder urothelial carcinoma (BLCA) demonstrated significant distributions in its NK (*p* = 0.1718), CD4+ (*p* = 0.2715), and macrophage (*p* = 0.3333) cell scores. Cholangiocarcinoma (CHOL) also showed high significance in its Tgammadelta (*p* = 0.6099), NK (*p* = 0.05944), and CD4+ (*p* = 0.6959) cell populations compared to BCC. Lymphoid neoplasm diffuse large B-cell lymphoma (DLBC) presented moderately significant *p*-values in scores for macrophages (*p* = 0.08976), Tgammadelta (*p* = 0.08334), and NK cells (*p* = 0.1221). Glioblastoma multiforme (GBM) also demonstrated high significance in its CD8+ (*p* = 0.2741), B cells (*p* = 0.2374), and Tgammadelta (*p* = 0.244) cell populations to BCC. Kidney renal clear cell carcinoma (KIRC) showed similar distributions for its NK (*p* = 0.2311), Th2 (*p* = 0.1368), and Tgammadelta (*p* = 0.2002) cell populations. Skin cutaneous melanoma (SKCM) had significantly similar cell populations of MSCs (*p* = 0.3872), NK cells (*p* = 0.05057), and CD4+ cells (*p* = 0.5715) to BCC. Finally, thyroid carcinoma (THCA) showed significance in its Tgammadelta (*p* = 0.7752), CD4+ (*p* = 0.4649), and NK (*p* = 0.0512) cell populations compared to BCC. 

Given that seven cancers had three similarly distributed features compared to BCC (BLCA, CHOL, DLBC, GBM, KIRC, SKCM, THCA), we favoured one of the three potential close relatives over the others. SKCM and BCC both originate in the skin and are UV-driven. A summary of the top three most similar cancers by cell type population is presented in Table 1, with an expanded version in Appendix A. Nonlinear dimensional reduction (tSNE) was used to visualize clustering results, using a perplexity of 50, depicted in Figure 3. Uveal melanoma (UVM) and thyroid carcinomas (THCAs) cluster in close spatial proximity.

## 4. Discussion

Similar to advanced BCC, kidney tumours such as KICH carry a heterogeneous microenvironment comprising both malignant and solid stromal cells, and a particularly elevated number of macrophages, and a Th2 cytokine shift. In particular, tumour-associated macrophages (TAMs), such as M2 macrophages, promote tumour growth through angiogenesis, which is enhanced when Th2-associated cytokines such as IL-4 and IL-10 are upregulated. Angiogenesis consists of the formation of new capillaries following ECM degradation, aided by matrix metalloproteinases (MMPs) and urokinase-like plasminogen activators (uPas) and their regulators. This process provides the tumour with oxygen, nutrients, and a pathway into circulation to metastasize [19]. Another similarity with BCC is that only a very low percentage of kidney chromophobe metastasizes: some studies estimate the risk of distant spread in KICH as low as 1.3%, compared to 0.0028–0.55% for BCC (average 0.04%) [20,21,22]. Thirdly, KICH differentiates itself from other kidney tumour cancers through its high mutation rate in the *TERT* gene promoter, in *PTEN*, and in *TP53* [23]. *TP53* is the second most commonly mutated gene in BCC after Sonic hedgehog alterations such as PTCH1 loss of function [24]. Moreover, studies focusing on mutations in the *TERT* promoter have been gaining much traction, with over 50% of BCC samples harboring the genomic aberration, specifically with C > T or CC > TT changes, distinctive for UV exposure [25,26,27]. BCC has the highest mutational burden of any human cancer, in contrast to KICH which has a very low mutational burden [24,28].

Currently, there are no existing precision oncology therapies for metastatic kidney chromophobe cancers due to its rarity and lack of genomic data [29]. However, it is particularly interesting to observe how both advanced BCC and KICH show contributions from tumour-associated macrophages (TAMs) and elevated Th1/Th2 immune cytokine scores [13,19]. Potential targeted therapies to reprogram or eliminate TAMs in KICH may also apply to advanced BCC [19]. In the future, the relationships between large amounts of TAMs, low metastatic rate, and mutations in *TP53* and *TERT* promoter should be further examined. 

Next on the list, sarcoma (SARC) was selected as the second BCC “relative”, specifically regarding its total lymphocytes, Tgammadelta, NK, and CD4 cells scores. Sarcomas have over 80 subtypes and comprise malignant tumours concerning non-epithelial connective tissue [30]. Molecular findings across various types of sarcomas are quite heterogeneous [31]. For most subtypes, first-line management includes a combination of surgery and radiation therapy, the latter either used adjuvantly and/or neoadjuvantly. Some sarcomas, such as dermatofibrosarcoma protuberans (DFSP), have actionable targets. In the case of DFSP, imatinib and other tyrosine kinase inhibitors can alter the overexpressed COL1A1-PDGFBB fusion proteins resulting in constitutive activation of PDGFR signaling [32]. Unfortunately, 25–50% of sarcoma patients still develop recurrent and/or metastatic disease after the surgery, which has prompted research to be focused on immunotherapy as a future treatment, in the form of immune checkpoint inhibitors, adoptive T cell transfer, or cancer vaccine [30].

Another common subtype of soft tissue sarcoma, myxofibrosarcoma, could be of great interest to BCC given the peculiar fibromyxoid stroma characterizing BCC [33]. For BCC, this stroma is believed to protect tumour cells and play a role in cancer promotion [34]. Myxofibrosarcoma tumour cells display complex karyotypes without a clear recurrent structural variant and molecular events are still poorly understood. However, the most frequent structural variation in this sarcoma is loss of chromosome arm 13q, which leads to the loss of well-characterized tumour suppressor RB in sarcomatous cells [35]. Circumventing dysfunctional Rb is a potentially actionable target [36].

The third “relative” from this analysis is skin cutaneous melanoma (SKCM), not surprising given that cutaneous melanoma is also a skin cancer and shares similar risk factors to BCC, such as ultraviolet radiation, fair skin complexion, familial history of skin cancer, genetic susceptibility, and numbers of nevi (acquired common and atypical nevi). Driver mutations of SKCM include mutations in BRAF, NRAS, and KIT oncogenes, and in NF1, PTEN, and TP53 tumour suppressors, among others [37]. Some driver mutations are sex-specific, such as DDX3X in males [15]. The current standard of treatment is surgical resection, followed by adjuvant treatment with immune checkpoint inhibitors (ICIs) or targeted therapy drugs, depending on the tumour staging [38]. It has been shown that *BRAF* mutations are often absent in non-melanoma skin cancers [39]; however, other less-frequently occurring mutations may provide insight on BCC. Mutations in *KIT*, a known tumour marker and proto-oncogene, accounts for about 3% of all melanomas, but are more prevalent in melanomas on chronically sun-damaged skin [40]. There have been conflicting results on whether c-KIT is implicated in BCC tumorigenesis, some in support [41,42,43], and others proving otherwise [44,45]. Despite this, the use of tyrosine inhibitors on BCC remains unfavourable, especially since its use in AML patients mainly caused BCC as a secondary cancer [46]. A more promising, yet under-researched, target may be JNK. JNK signaling is highly responsive to UV radiation and inflammatory cytokines, among others. Its downstream effects include upregulating AP1, a JNK effector responsible for extracellular matrix (ECM) remodeling [47]. In melanoma, JNK has been found to be implicated in melanoma growth and progression, initiated through activation by IL-1β, whose receptors are highly expressed on Th2 cells [48,49]. Concerning BCC, it has been reported that JNK is a master mediator for numerous key pathways, such as Wnt, Shh, and YAP [50]. In addition, the aforementioned involvement in the *TERT* promoter gene, which is mutated in over 70% of melanoma cases, and over 50% in BCC, may be of interest as a therapeutic target [51]. *TERT* mutations have been shown to be positively correlated with a higher tumour mutational burden (TMB) value, neoantigen load, and tumour purity. Its expression as a tumour-associated antigen has primed it to be an ideal target of immunotherapy, with numerous studies and trials currently in development and underway [52]. As such, along with its crucial role in melanoma resistance, more studies on JNK and *TERT* promoter in BCC may uncover more information on advanced and treatment-resistant BCC. 

Omics profiling has yielded important clues into tumour initiation, promotion, and progression which resulted in development of targeted therapies [53]. For BCC, Sonic hedgehog inhibitors that selectively inhibit SMO include vismodegib (Erivedge) [8] and sonidegib (Odomzo) [54]. Vismodegib has been shown to lead to the recruitment of cytotoxic T cells into the tumour microenvironment, with concurrent upregulation of type I MHC in BCC tumour cells [55]. Treatment of SMO inhibitors leads to at least a partial response in ~2/3 of advanced BCC patients, but >25% of patients discontinue therapy due to frequent, hard-to-tolerate side effects such as alopecia, dysgeusia, and muscle spasms [9]. BCC tumour cells often acquire resistance to SMO inhibitors, which may result from a cell identity switch towards a mesenchymal-stem-cell-like phenotype [56]. Despite its very high tumour mutational burden, anti-PD-1 therapy such as cemiplimab fails in ~70% of advanced BCC patients [57].

## 5. Conclusions

We have identified three potential BCC “relatives” sharing similar tumour microenvironment findings. We also reviewed underlying literature for current management therapies and potential novel approaches in these three relatives which may apply to advanced BCC patients. Kidney chromophobe particularly stood out as a promising BCC “relative”, demonstrating strong similarities in TAM and Th2 cytokine profiles. Targeting the RB pathway in myxofibrosarcoma may be of interest regarding modulating the fibromyxoid stroma of BCC, rendering it less suitable to promote BCC growth. Exploring lesser-known contributors to melanoma, such as c-KIT and JNK, may also be of interest. Although there still does not exist a definitive treatment for advanced BCC, new targeted therapeutics used in other malignancies with closely related microenvironment changes might provide novel management strategies to be studied and potentially employed in the future.

## Figures and Tables

**Figure 1 cancers-15-00305-f001:**
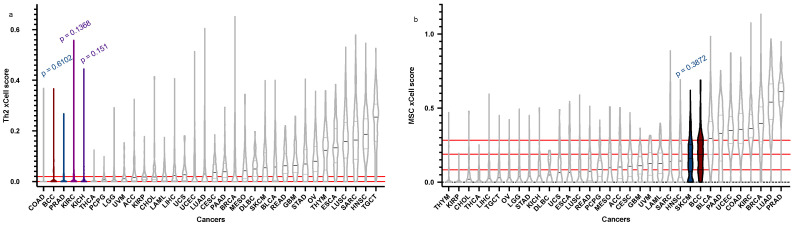
BCC vs. BCC “relatives”: xCell. In silico cell type enrichment scores obtained using xCell, comparing BCC (dark red) to other BCC “relatives” (white), with statistically similar cancers denoted (grey), with the top three in colour (dark blue, purple, violet) with their corresponding *p*-values above. Truncated violin plots of Th2 immune cytokine (**a**), and mesenchymal stem cells (MSCs) (**b**) are shown in order of ascending median. BCC median, 1st quartile, and 3rd quartile are delineated in red (where applicable).

**Figure 2 cancers-15-00305-f002:**
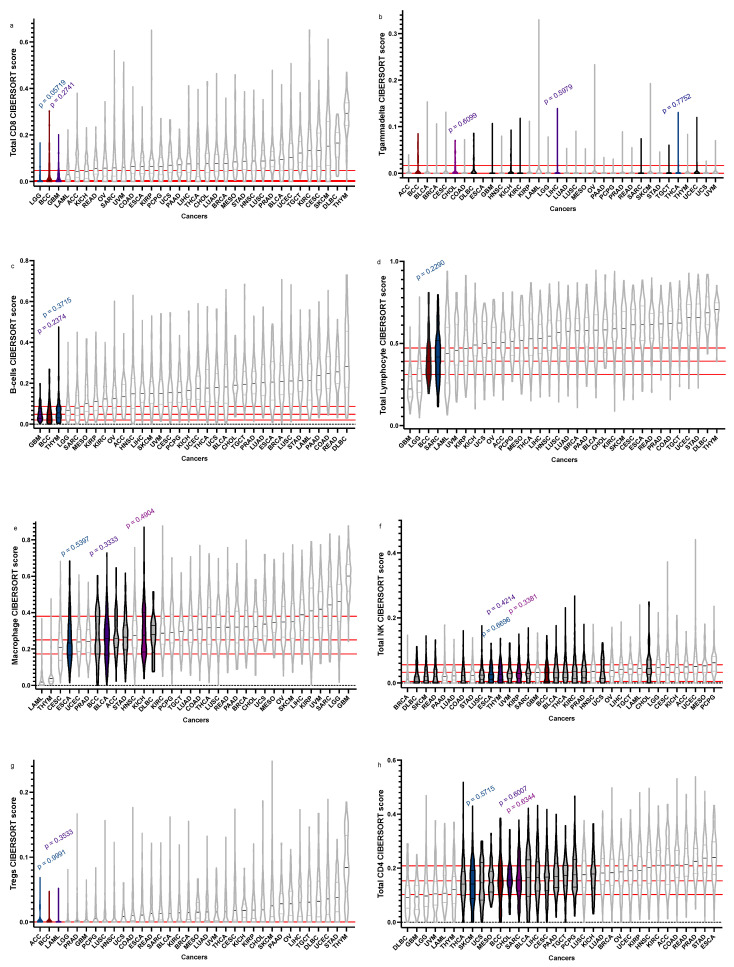
BCC vs. BCC “relatives”: CIBERSORTx. In silico immune cell fractions obtained using CIBERSORTx, comparing BCC (dark red) to other BCC “relatives” (white), with statistically similar cancers denoted (grey), with the top three in colour (dark blue, purple, violet) with their corresponding *p*-values above. Truncated violin plots of total CD8 T cells (**a**), T gamma delta cells (**b**), total B-cells (**c**), total lymphocytes (**d**), total macrophages (**e**), total NK cells (**f**), regulatory T cells (Treg) (**g**), and total CD4+ cells (**h**) are shown in order of ascending median. BCC median, 1st quartile, and 3rd quartile are delineated in red (where applicable).

**Figure 3 cancers-15-00305-f003:**
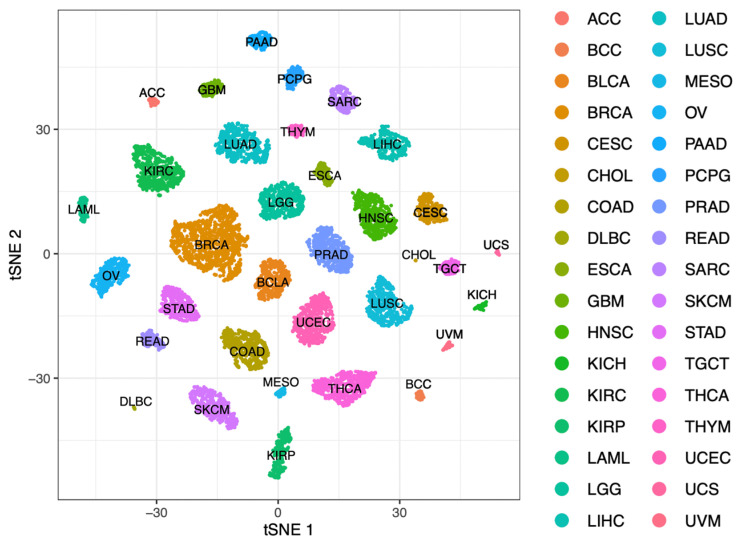
*t*-distributed stochastic neighbour embedding plot of the tumour microenvironment scores (from CIBERSORTx and xCell) of 33 TGCA cancers with BCC, coloured by cancer type.

**Table 1 cancers-15-00305-t001:** Summary of the top three significant (if available) closest BCC “relatives” by cell type.

Cell Type	BCC Score (Median), Q1, Q3	BCC “Relatives” Score (Median)	Q1	Q3	*p*-Value
Th2	4.188 × 10^−18^, 0, 0.01980129	PRAD = 1.07 × 10−17	0	0.01706565	0.6102
		KICH = 0.001056705	0	0.007381045	0.151
		KIRC = 2.23 × 10−17	0	0.03051409	0.1368
MSC	0.188433346, 0.08336783, 0.28205374	SKCM = 0.16096867	0.07202425	0.25446426	0.3872
Total CD8+	0.00445073, 0, 0.043739228	GBM = 0.01048155	0	0.03021879	0.2741
		LGG = 0	0	0.01996548	0.05719
Tgd	0, 0, 0.008231755	THCA = 0	0	0.008063912	0.7752
		CHOL = 0	0	0.01282003	0.6099
		LIHC = 0	0	0.006720286	0.5979
B-cells	0.04857795, 0.020757612, 0.086950544	THYM = 0.055183104	0.03108108	0.08455359	0.3715
		GBM = 0.04815724	0.028318876	0.075344991	0.2374
Total Lymphocytes	0.39140159, 0.3132104, 0.4688854	SARC = 0.416998654	0.3124302	0.5153758	0.2290
Macrophages	0.25033106, 0.1763854, 0.3784326	ESCA = 0.2331247	0.1675734	0.3352133	0.5397
		KICH = 0.276729	0.1843325	0.3949821	0.4904
		BLCA = 0.2590653	0.16693727	0.33762817	0.3333
Total NK	0.0325587, 0.005046464, 0.053041073	THYM = 0.02682497	0.009408674	0.049124771	0.6696
		ESCA = 0.02523973	0.01152386	0.04604726	0.4214
		KIRP = 0.02878749	0.01382830	0.04874396	0.3381
Tregs	0, 0, 0	ACC = 0	0	0	0.9991
		LAML = 0	0	0.00315034	0.3533
Total CD4+	0.15296126, 0.1031139, 0.2082608	CHOL = 0.156189	0.1321129	0.1952696	0.6959
		SARC = 0.1562586	0.1154291	0.2083701	0.6334
		SKCM = 1412604	0.08728384	0.19168763	0.5715

## Data Availability

Previously published whole-genome RNA-Seq data originated from: Atwood et al. (GEO accession number GSE58377) [58], Bonilla et al. (EGA accession number EGAS00001001540) [59], and Sharpe et al. (EGA accession number EGAS00001000845) [60]. The pan-cancer datasets analyzed are available from The Cancer Genome Atlas (TCGA) database (https://portal.gdc.cancer.gov/ accessed on 19 June 2019).

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
