# Peer review of "Comparison of the Basal Cell Carcinoma (BCC) Tumour Microenvironment to Other Solid Malignancies"

_cancers, 2023, doi:10.3390/cancers15010305_

Round 1

Reviewer 1 Report

Thanks for this work on finding common links, cell types between cutaneous BCCs and an array of other tumors. 

A large number of possible tumors are studied. Very few were found related. Information regarding that those that do not resemble BCC could be reduced. For example, Table 1 shows an array of data that is not statistically significant. This could be summarised.

In contrast, the authors identified three tumors with prominent links, including melanoma, - also a largely cutaneous origin tumor. Greater detail on these links would be helpul, whilst reducing the content on non significant findings.

There are times when information is included in the incorrect section of the manuscript.

For example, the first few sentences of 'Results' are in fact background, not results. 

Further, Conclusion starts with background information, past work by the research team and commenting on likt search is 'introduction' and 'methods' , not conclusion. The authors need to refine the conclusion section to solely the conclusions of this specific study. 

Author Response

Reviewer #1

Thanks for this work on finding common links, cell types between cutaneous BCCs and an array of other tumors. 

R1-C1

A large number of possible tumors are studied. Very few were found related. Information regarding that those that do not resemble BCC could be reduced. For example, Table 1 shows an array of data that is not statistically significant. This could be summarised.

Our response: The data listed in Table 1 presents cancer types with statistical significance in their corresponding immune cell type compared to BCC, with the top three most significant (by highest p-value) in bold. In this type of analysis, we aimed to find most similar distributions, hence those that will not be rejected by a goodness of fit test. We do agree that this table can be summarized to only show the top three (if available) similar cancers, with the rest displayed in Supplementary Figure 1.

R1-C2

In contrast, the authors identified three tumors with prominent links, including melanoma, - also a largely cutaneous origin tumor. Greater detail on these links would be helpful, whilst reducing the content on non-significant findings.

Our response: We really appreciate the insightful suggestion from Reviewer #1. As such, modifications have been made, namely in the KICH and SKCM sections of the discussion, by expanding on the literature covering TERT promoter mutations.

Moreover, studies focusing on mutations in the TERT promoter have been gaining much traction, with over 50% of BCC samples harboring the genomic aberration, specifically with C>T or CC>TT changes, distinctive for UV-exposure [25-27]. Lines 549-552

In addition, the aforementioned involvement in the TERT promoter gene, which is mutated in over 70% of melanoma cases, and over 50% in BCC, may be of interest as a therapeutic target [51]. TERT mutations have been shown to be positively correlated with a higher tumor mutational burden (TMB) value, neoantigen load, and tumor purity. Its expression as a tumour-associated antigen has primed it to be an ideal target of immunotherapy, with numerous studies and trials currently in development and underway [52]. Lines 655-661

R1-C3

There are times when information is included in the incorrect section of the manuscript.

For example, the first few sentences of 'Results' are in fact background, not results. 

Further, Conclusion starts with background information, past work by the research team and commenting on lit search is 'introduction' and 'methods', not conclusion. The authors need to refine the conclusion section to solely the conclusions of this specific study. 

Our response: We agree that the first sentences of “Results” serve more as background information and has therefore been removed.

Reviewer 2 Report

In the provided manuscript, the main question is to find other cancers with similar microenvironment featuress to that of BCC. This topic is relevant as there are no reliableexperimentalallmodelsin either animal or cell lines for BCC. Finding BCC “relatives” is essential for developing new drugs for more aggressive subtypes of BCC.

  1.  I want to ask the authors if any similar studies have been done previously or if this is the first study of this kind. 

  2. Is a sample size of 75 typical for other genomic studies?

The conclusions are well-prepared and are supported by the results. The references are appropriate. especially enjoyed the figures, they are very nicely prepared. I added the pdf with small note to consider.

Author Response

Reviewer #2

In the provided manuscript, the main question is to find other cancers with similar microenvironment features to that of BCC. This topic is relevant as there are no reliable experimental models in either animal or cell lines for BCC. Finding BCC “relatives” is essential for developing new drugs for more aggressive subtypes of BCC.

R2-C1

I want to ask the authors if any similar studies have been done previously or if this is the first study of this kind. 

Our response: Comparison of immune cell types between TGCA cancer subtypes have been previously reported (https://doi.org/10.1016/j.immuni.2018.03.023, https://doi.org/10.1038/s41598-020-66449-0), as well comparing BCC to other TGCA cancer types (https://doi.org/10.3389/fonc.2021.752579) regarding mutations and CNVs, but not related to TME. However, to our knowledge, the comparison of a non-TGCA cancer TME to other TGCA cancers has not yet been reported and hence is the first study of this kind.

R2-C2

Is a sample size of 75 typical for other genomic studies?

Our response: Sample size is often determined by the nature of the available data. Although it is always preferable to have a larger sample size, our previous studies (doi: 10.1007/s12079-020-00563-6., doi: 10.26508/lsa.202000651) have detected statistical significance for various research hypotheses using the same 75 samples. There is a paucity of genomics data for BCC.

R2-C3

The conclusions are well-prepared and are supported by the results. The references are appropriate. especially enjoyed the figures, they are very nicely prepared. I added the pdf with small note to consider.

Our response: We agree with the suggested note and have expanded the introduction to include the citations mentioned.

  1. Chlebicka I, Stefaniak A, Matusiak Ł, Szepietowski JC. Basal cell carcinoma: what new can be learned about the most com-mon human cancer? A cross-sectional prospective study of 180 cases in a single centre. Advances in Dermatology and Aller-gology/Postępy Dermatologii i Alergologii. 2021;38(6):1086-1091. doi:10.5114/ada.2021.106026.
  2. Marzuka AG, Book SE. Basal cell carcinoma: pathogenesis, epidemiology, clinical features, diagnosis, histopathology, and management. Yale J Biol Med. 2015;88(2):167-179. Published 2015 Jun 1.

Round 2

Reviewer 1 Report

Thanks for refinements to this piece.

I feel it is improved. 

My only ongoing concern is that it is acting as two manuscripts:

1) A research study

2) A literature review

Any further development would be best focusing on the fact that this manuscript is (presumably) focues on (1).